# The internet of things deployed for occupational health and safety purposes: A qualitative study of opportunities and ethical issues

Maeva El Bouchikhi[1], Sophie Weerts[1], Christine Clavien[2]*

1 Swiss Graduate School of Public Administration (IDHEAP), University of Lausanne, Lausanne, Switzerland, 2 Institute for Ethics, History, and the Humanities (iEH2), University of Geneva, Geneva, Switzerland

* Christine.clavien@unige.ch

## Abstract

The deployment of the Internet of Things (IoT) technology (connected devices enabling algorithmic analysis of behaviour and individualized feedback) has been growing increasingly over the last decades, including in the workplace where they can serve occupational safety and health (OSH) purposes. However, although the IoT is deployed for good aims, the use of these devices raises numerous ethical issues which have had little literature specifically dedicated to them. To fill this gap, we have investigated the ethical views of key stakeholders on the deployment of IoT for OSH. We conducted a focus group and semi-structured interviews including 24 stakeholders and analysed transcripts with an open coding method. Participants were favourably disposed towards the use of some versions of IoT (posture-tracker chair, step-tracker watch), but rejected other devices (sound-tracker on computer). They highlighted an important number of ethical issues which could be grouped into five overarching categories: goal relevance, adverse side effects, role of employees, data process, and vagueness. Their inputs were remarkably coherent with the issues highlighted in the academic literature. They also felt quite disenchanted and shed a stark light on the lack of information at the disposal of stakeholders in the front line to assess such technology. Our results provide important ground material on which to base necessary and still-awaited guidelines and regulation instruments.

## Introduction

Computers, robots, drones, cameras or integrated systems are increasingly becoming part of our working environments and are organized in networks, constituting the Internet of Things (IoT). This unprecedented deployment of interrelated information technologies in the 21st century has improved productivity in many sectors and contributed to major transformations of working conditions, sometimes leading to intensive algorithmic management and sectoral replacements of workers by machines [1–4]. Optimistic authors announce an "age

**Data availability statement:** All relevant data are contained within the manuscript and sup-porting information files. Further information including informed consent form and the full book of codes is available on osf.io/e97v2. Some of our participants are public figures and some inputs during the discussions provide sensitive information about current practices in recognizable local companies. Due to the risk of de-anonymisation, we committed not to make transcripts publicly available in the data repository.

**Funding:** SW and CC obtrainded funding for this research from the Swiss National Science Foundation (grant no. 187429) within the Swiss National Research Programme (NRP77) on "Digital Transformation" (https://www.snf.ch/en/hRMuYd5Qqjpl1goQ/page/researchin-Focus/nrp/nrp77). The SNSF is a nonprofit national funding agency that was not involved in any step of this study. The authors have no financial or competitive interests to declare. The funders had no role in study design, data collection and analysis, decision to publish, or preparation of the manuscript. The authors have no financial to declare.

**Competing interests:** The authors have declared that no competing interests exist.

of augmentation", where humans and machines co-evolve harmoniously in an integrated manner [5]. Whether positive or negative, such technologized work environment is made possible by the deployment of smart devices, called Internet of Things devices (IoTs). They are equipped with information sensors and connected through Internet-based communication networks. They can collect data *en masse* on work production, individual characteristics, and group dynamics. This data enables the analysis of workflows with unprecedented granularity, leading to reliable predictions and preventive and adaptive measures.

IoTs can also serve occupational safety and health (OSH) purposes. They include porta-ble cameras, augmented or virtual reality tools, wearables, apps, smart clothes, or protective equipment such as exoskeletons [6, 7]. In a proactive manner, these devices are used to assess risks and prevent work-related illnesses or safety incidents. In a reactive manner, they are used to improve the quality of the investigation and reporting of adverse events or to reduce their negative effects [8].

Despite the value of IoTs for OSH purposes, their deployment may involve worrying drawbacks and raise ethical issues that are overlooked by relevant stakeholders. Policymakers and stakeholders implicated in the implementation of IoTs in working contexts are ambivalent regarding ethical issues. Some warn of the risks posed by IoTs, while others are extremely opti-mistic about the value of the technology [9–11]. For years, scholars in various fields (including the social sciences, law, medical ethics, business ethics, and ethics of new technologies) have pointed out worrying issues, such as unexpected adverse effects on the mental and social well-being of workers, or cases of problematic dual use when IoT for OSH also collect data for surveillance purposes or performance analysis [12–20]. These many relevant discussions are somewhat scattered over various academic fields and interwoven with related but broader top-ics of discussion, such as the social impact of AI, algorithmic management in a work context, or self-quantification of health [21]. This literature, however, clearly points out the need to assess the technology ahead of its deployment and prevent possible abuses and questionable uses.

At the empirical level, there is also interesting burgeoning work on stakeholders' evalua-tion of IoTs. In a survey addressed to 702 European managers, Pütz and colleagues [22] found significant acceptance of IoT devices specifically designed for monitoring mental health. Nonetheless, positive expectations were counterbalanced by privacy concerns. In a qualita-tive study, Bovens and colleagues [23] investigated personnel's view on the values relevant to deploying an occupational health management IoT. Their participants allotted importance to the values of security, trust and hierarchy, but did not report a strong need for ethics support. In another study, Le Feber et al. [24] explored the ethical and privacy issues that may occur with the introduction of sensors in the workplace with 32 stakeholders. The stakeholders were open-minded to the use of sensors for applied purposes such as OSH. Participants expressed the following: end users should be involved in the decisions on the purpose and application of IoT, possible negative adverse effects need to be considered, and continuous sensor-based monitoring is too intrusive. While important, these results nevertheless remain limited and difficult to generalise. Notably, the Bovens et al. study involved a very specific category of participants: reservists who had previously accepted use of such a device. It is unclear to what extent participants' evaluation was influenced by the social values of their profession and by an *a priori* positive attitude towards IoTs. And to frame the interview discussion, Le Feber et al used an ethical grid that included the values of fairness, proportionality, and intrusiveness. It is therefore unclear whether participants would have come up with the same ethical themes without being explicitly primed.

In this context, we consider that there is still a need for a more encompassing work in the area of the ethics of IoTs. Firstly, multidisciplinary and fragmented ethically relevant informa-tion and reflections need to be collated in a comprehensive overview. Secondly, more data is

needed on the points of view of stakeholders, including company managers, OSH specialists, and first-hand users (employees). It currently remains unclear which ethical issues are relevant to them, and to what extent they are aware of the relevant benefits and risks associated with the use of these devices in working contexts.

Relevant ethical insights on IoTs can be drawn from academic literature and from stakeholders' point of view. In the context of a larger research project aiming to provide a comprehensive overview of the ethical opportunities and problematic issues posed by *IoTs for OSH purposes*, we used a mixed method approach, conducting a scoping literature review (reported in a separate paper [21]), and a complementary empirical investigation involving stakeholders (presented here). This double methodological perspective enables to identify ethical concerns experienced by stakeholders that have received limited or no attention in the existing literature (business ethics, digital ethics and health ethics), and, reversely, to highlight issues that are discussed in the literature, but systematically ignored by stakeholders due either to a lack of awareness or to an underestimation of their significance.

The literature review [21] revealed an unexpectedly detailed and substantial list of issues posed by using IoTs for OSH purposes, including the topics of surveillance and problematic data re-purposing, the risk of suboptimal data management, and the difficulty of informing, consulting, and obtaining consent from employees. The review also revealed an important number of possible unintended and unpredictable adverse effects (e.g. weakening of workers' competencies and self-confidence, exacerbation of social stigma and inequalities, deresponsibilization of employers), in addition to contextual factors that enhance risks and ethical difficulties (e.g. blurred limits between professional and private life, complexities and lack of competencies, costs of ethical safeguards).

This paper presents the results of the empirical investigation involving one focus group and a series of qualitative interviews. The goal was to investigate the opportunities, challenges, and ethical issues, as viewed by a large variety of stakeholders in possession of firsthand experience with IoTs for OSH purposes (workers, worker representatives, policymakers, managers, OSH specialists, occupational physicians, and critical analysts). For the sake of concision and relevance, we limited our research on IoTs deployed for the purpose of preventing health hazards (proactive approach). These devices are the easiest and most likely to be implemented by large and small-scale companies and have received little attention in the ethics literature. We analysed our data with a grounded theory method and conducted the literature review mentioned above at a later stage to minimize the theoretical influence of academic discussions on the empirical analysis presented here. In the discussion section, we compare and discuss the outputs of both lines of research.

Overall, our results help to structure the debate on the ethical issues at stake. We provide fundamental knowledge for the critical assessment of IoTs deployed for OSH purposes, and for the development of ethical frameworks, guidelines, regulation instruments, safeguards, and accompanying measures to mitigate individual and societal risks. We also highlight ethical issues that are likely to be overlooked either in the academic realm, or among stakeholders and decision-makers. Our results are thus helpful for theoreticians and policymakers.

## Materials and method

We conducted one focus group and semi-structured individual interviews with experts (employee with user experience, advocate of employees, company manager, Human Resources manager, occupational physician, medical information physician, cantonal physician, occupational safety and health professional, public administrators, health insurance representative, IT developer, critical analyst, tech journalist, lawyer). We chose this double

method because focus groups usually enable debate, discussion, sharing of ideas, and generate more refined lines of thought, while individual interviews provide more space for shyer participants to speak, especially for participants from lower down in the hierarchy. Interviews also allow to adapt the language (French, English) to participants' ease.

## Ethics statement

The protocol was approved by the "Commission Universitaire pour une Recherche Ethique à l'Université de Genève" (Req-ID: CUREG-2021-11-124; date of acceptance: 03.12.2021). Willing participants were informed of the study aims and procedures and signed a written informed consent form. Participants' recorded responses were transcribed. We then anonymised all data before destroying the records. This study is preregistered on OSF registries: doi. org/10.17605/OSF.IO/4JV9G.

## Material

We constructed our interview grid in English and French (all our coders are bilingual), with the aim of capturing a broad range of perspectives on the benefits, challenges, and risks associated with IoT devices for occupational health that could plausibly or that were already used in Switzerland. For this, we developed three fictional but plausible scenarios (see details in S1 Text) of types of devices that are already being used across the globe: the deployment of connected chairs in the workplace to help employees avoid and correct bad posture (hereafter, Posture-tracker Scen); the use of smart-watches in the context of a corporate wellness program designed to motivate employees to make more daily steps (Step-tracker Scen); the deployment of sound trackers in employees' offices and homes to detect and reduce stress level (Sound-trackers Scen). In order to cover as many relevant situations as possible, our three scenarios diverged on the following parameters: type of device used (more or less physically perceptible, more or less used in everyday-life), health purpose of the device (to prevent physical *versus* mental illness), time of implementation of the device (deployed over a short *versus* long period in the workplace), mode of decision for implementing the device (after collective consultation with employees, after consultation with employees representatives, without consultation), extension of activity of the device (workplace only *versus* extension in employee's private sphere), type of data collected (postural data, personal data, health data, behavioural data, speech tone, etc.), type of data storage (in-house *versus* outsourced to the device's provider), mode of data analysis (done in-house *versus* outsourced to the device's provider), data accessibility (to employees only *versus* to physician or Human Resources [HR] or direct managers). To confirm that our scenarios were realistic, and to obtain a first gut feeling appreciation of participants, the interview grid included three starting closed questions: "Could this scenario be realistically used in Switzerland?"; "Do you know Swiss companies using similar IoT technology (provide a yes/no answer–no need to tell names) And how many companies?"; "Could this scenario be legally used in Switzerland ?" (the latter question was added in the course of the study, thus we do not have answers from all participants); "Would you (as employer versus as employee) be in favour of implementing this solution?". We then proceed with an open question: "In your view, what are the chances (opportunities, advantages) and the ethical issues (possible risks, topic of controversy) related to this scenario?". In addition, we developed follow-up questions to be used in case participants did not spontaneously discuss some issues that we considered relevant, in particular issues related to data flow: for each scenario, we showed to participants a map of the data flow process (see Supporting information 1) and asked whether they saw new or further issues related to that process. We also asked additional questions on variations on the scenarios, such as "What if

the data analysis was done in-house by the HR unit of the company?" (Posture-tracker Scen & Sound-trakers Scen) or "What if the individual data reports were transmitted to the occupational physician or to the HR service?" (Step-tracker Scen). In the sound-tracker scenario, we asked participants for their views on the role of the local manager, and "What if the computers are also equipped with a camera for capturing facial expressions and eye movement?" At the end of the interview, we added a global follow-up question on the decisions procedure ahead of the deployment of the device, and the role employees should take in that process: should employees or their representatives be consulted or can the decision be taken in a top-down manner?

## Participant recruitment

To map as many ethical issues as possible and identify different points of view, following a maximum variation sample method [25], our goal was to include a broad range of actors in different domains of activity related to IoTs (see detail in Table 1): we aimed at covering knowledge and competencies related to 1) the development and production of IoTs (e.g. IT developer, manager in a start-up producing IoTs), 2) the deployment and use of IoTs in private companies (e.g. private company manager, Human Resources manager, OSH professional, occupational physician, end-user employees, advocate of employees, health insurance representative), and 3) the assessment, control and regulation of such use (e.g. lawyer, public administrator, cantonal physician, tech journalist, critical analyst working in a university or think tank). Overall, we also aimed at a diversity in gender, hierarchical level on the workplace, socio-cultural regions in Switzerland, and a variety of public and private working sectors (e.g. university, high tech start-up, federal administration, think-tank, media sector, goods-producing company, selling company, politics). Given that Switzerland is a small country and given the specificity of the type of participants we required, we did not use a random sampling method but invited stakeholders based on personal recommendations or online searches of profiles.

## Data collection

The focus group took place during a workshop in December 10, 2021. The individual interviews were conducted from January 4 to April 29, 2022. To obtain the richest data we recorded a focus group and further individual online interviews until reaching data saturation (no new ethical issues or benefits were mentioned by additional participants) *and* a maximum variation sampling (at least one participant in each category and two participants in the most relevant categories).

We used the same method for the focus group and individual interviews. At least one day before the event, participants received an information form explaining the general aim of the project. On the day, participants received the consent form. They signed it before we began recording. Based on an interview grid (S1 Text), the moderator read the first scenario, answered any comprehension questions, and asked a series of closed and open questions in sequence (during the focus group, each participant was asked in turn to answer each question, and once this was done, the floor was open to free discussion). When there was no longer any novel input, the moderator proceeded to the next question. In total, three scenarios were presented to participants with the same type of questions. If participants did not spontaneously discuss all topics we had identified as relevant, the moderator asked some follow-up questions, and showed figures of the data flow in relation to each scenario, making it more explicit with whom data was shared, where it was stored, and who made the analysis (S1 Text). All scenarios were available in French and English, and participants could choose between

**Table 1. Characteristics of study participants.**

| Participant ID * | Category of participant | Gender | Age category | Language used | Hierarchical level at work | Working for a Public/Private Organisation |
|---|---|---|---|---|---|---|
| IIp1 | IT developer | M | 50–60 | French | Manager | Private |
| IIp2 | Health professional | F | 40–50 | French | Manager | Private |
| IIp3 | Health Insurance & Critical analyst | F | 20–30 | English | Employee | Public |
| IIp4 | IT developer | F | 40–50 | English | Manager | Public |
| IIp5 | Critical analyst | M | 30–40 | French | Manager | Public |
| IIp6 | Health professional | M | 40–50 | French | Manager | Private and Public |
| IIp7 | Decision-maker | F | 40–50 | French | Employee | Private |
| IIp8 | OSH professional | M | 30–40 | French | Manager | Private |
| IIp9 | Health professional & Health Insurance | M | 40–50 | French | Manager & Start-up | Private |
| IIp10 | IT developer | M | 40–50 | French | Manager & Start-up | Public and Private |
| IIp11 | Decision-maker | M | 40–50 | French | Manager | Private |
| IIp12 | Public administration | M | 40–50 | English | Employee | Public |
| IIp13 | IT developer | M | 50–60 | French | Manager | Private |
| IIp14 | Lawyer | M | 30–40 | French | Employee | Private |
| IIp15 | OSH professional & Employee with user experience | M | 40–50 | French | Employee & Manager | Private |
| IIp16 | Employee with user experience | F | 30–40 | French | Employee | Private |
| IIp17 | Critical analyst | M | 50–60 | French | Employee | Private |
| FGp18 | Critical analyst | M | 40–50 | French | Start-up | Private |
| FGp19 | Public administration | M | 30–40 | English | Manager | Public |
| FGp20 | Critical analyst & Health professional | M | 60–70 | Fr & Engl | Manager | Public |
| FGp21 | IT developer | M | 20–30 | English | Employee | Public |
| FGp22 | Critical analyst | M | 40–50 | Fr & Engl | Start-up | Public and Private |
| FGp23 | Decision-maker & Critical analyst | M | 30–40 | Fr & Engl | Manager | Private |
| FGp24 | Advocate of employees | M | 40–50 | Fr & Engl | Manager | Public & Private |

All participants have some personal experience with the use, development, or analysis of IoT technology for occupational health. Categories include: 2 employees with user experience, 1 advocate of employees (unionist and left wing politician), 3 decision-makers (1 private company manager, 2 Human Resources managers), 4 health professionals (1 occupational nurse, 1 occupational physician, 1 medical information physician, 1 cantonal physician), 2 occupational safety and health professionals, 2 public administrators (1 representative for Swiss Economic Affairs, 1 expert for the Swiss Health Promotion), 2 health insurance representatives (1 administrative, 1 researcher), 5 IT developers (1 engineer, 1 tech researcher and engineer; 3 developers of IoT solutions), 6 critical analysts (1 researcher in sociology, 1 researcher in clinical informatics, 1 member of a think-tank; 1 adviser on IoT technology, 1 ethicist, 1 tech journalist), 1 lawyer. Participants worked in different sociocultural regions in Switzerland: French-speaking = 14; German-speaking = 7; Italian-speaking = 1.

* IIp = Individual Interview participant; FGp = Focus Group participant.

both languages, including during the focus group (knowing that all participants had a passive understanding of the other language).

## Data analysis

Participants' answers to the closed questions are compiled in the descriptive Table 2. To leave space for unanticipated information to emerge when analysing participants' answers to open questions, we employed a method involving elements from grounded theory [26]. First, we transcribed the recording verbatim in the original language with help of the software Word, pack version Microsoft 365, and anonymized the data. Coding was done in English only. After an immersion phase of reading and note taking, we proceeded to the initial coding of the verbatim transcription using software (Atlas.ti). To guide this initial step, we used a "start

**Table 2. Summary of participants' responses to our closed questions.**

| Questions asked | Initial Yes or rather Yes | Initial No or rather No | No clear opinion | Change of opinion |
|---|---|---|---|---|
| **Could this scenario be realistically used in Switzerland?** | **Total: 54 (78%)** | **Total: 13 (19%)** | **Total: 2 (3%)** | **Total: 4 (6%)** |
| Posture-tracker Scenario | 17 | 4 | 0 | 0 |
| Step-tracker Scenario | 22 | 1 | 1 | 0 |
| Sound-tracker Scenario | 15 | 8 | 1 | from Yes to No: 2 from No to Yes: 1 |
| **Could this scenario be legally used in Switzerland?** | **Total: 39 (57%)** | **Total: 17 (25%)** | **Total: 13 (19%)** | **Total: 3 (4%)** |
| Posture-tracker Scenario | 15 | 2 | 4 | from Yes to Don't know:1 |
| Step-tracker Scenario | 17 | 4 | 3 | 0 |
| Sound-tracker Scenario | 7 | 11 | 6 | from Yes to No: 2 |
| **Do you know Swiss companies using similar or comparable IoT technology? Provide a yes/no answer. No need to tell names.** | **Total: 29 (42%)** | **Total: 31 (45%)** | **Total: 9 (13%)** | **Total: 3 (4%)** |
| Posture-tracker Scenario | 4 | 14 | 3 | from Yes to No : 1 from No to Yes : 1 |
| Step-tracker Scenario | 18 | 4 | 2 | from No to Yes : 1 |
| Sound-tracker Scenario | 7 | 13 | 4 | 0 |
| **Would you, *as a head of the company*, be in favour of [implementing this solution]?** | **Total: 38 (55%)** | **Total: 30 (43%)** | **Total: 1 (1%)** | **Total: 2 (3%)** |
| Posture-tracker Scenario | 18 | 3 | 0 | from Yes to No: 1 |
| Step-tracker Scenario | 18 | 5 | 1 | 0 |
| Sound-tracker Scenario | 2 | 22 | 0 | from No to rather Yes: 1 |
| **Would you, *as an employee*, be in favour of [implementing this solution]?** | **Total: 34 (49%)** | **Total: 35 (51%)** | **Total: 0 (0%)** | **Total: 0 (0%)** |
| Posture-tracker Scenario | 13 | 8 | 0 | 0 |
| Step-tracker Scenario | 17 | 7 | 0 | 0 |
| Sound-tracker Scenario | 4 | 20 | 0 | 0 |

"Yes or rather Yes" (versus No) means that overall the participant responded positively (versus negatively) to the question. The column "Change of opinion" reports the number of participants who changed their evaluation later during the interview or focus group because they realized that they had misinterpreted the question or because additional thoughts made them change their mind. Note that 3 participants in the focus group missed the discussion of the chair scenario, which explains why not all total 24.

list" of codes (applied top-down), containing a limited number of broad categories classically used in value maps (autonomy, justice, safety, security, solidarity, etc.), goods (health, well-being), harms (dependency, exploitation), and concerns (for privacy, for well-being, etc.). All other first-level codes were generated bottom-up. While proceeding, items from the start list were modified, expanded, or deleted until we obtained a unified (top-down and bottom-up) list of codes. We then focused on codes only and searched for emerging themes and sub-themes, using word clouds (Atlat.ti function) and mindmaps. We renamed, merged, split or deleted codes whenever necessary, until we obtained a clear branching organization around a low number of overarching categories (final conceptual framework). While proceeding, we systematically checked categories, themes and first order codes matched with the verbatim transcription. All steps were mainly carried out by the first author. In addition, 50% of the initial transcripts were double coded independently by the second and last authors and

an independent colleague, then compared and discussed. 80% of the thematic organization of codes was done or reviewed in closed feedback loops during regular meetings between the first and last authors. The second author occasionally checked overall consistency, and matching between categories, themes, and verbatim transcription, to ensure that concepts were clearly defined, appropriately derived from the data, and that codes and themes were being used consistently.

**Two-step analysis (with and without answers to follow-up questions).** Since we asked follow-up questions designed to attract participants' attention to relevant ethical issues, overall responses were biased towards the topics of our follow-up questions. To account for this, we separately analysed participants' responses before the follow-up questions. In a second stage, we added follow-up responses to the data set and observed which additional codes emerged. This enabled us to report participants' responses with and without additional guiding questions.

**Subgroup analyses.** To observe whether different types of participants were more (or less) sensitive to specific ethical issues, in a post-hoc analysis, we created four subgroups of the dataset: *women*, *men*, participants *with a conflict of interest* in favour of the technology (decision-makers, health insurance representatives, IT developers), and participants *without a conflict of interest* in favour of the technology (employee with user experience, health professionals, occupational safety professionals, critical analysts, public administrators, lawyers). We only kept codes related to discussions of ethical issues (e.g. removed codes related to the benefits of the technology, to the condition of acceptability of the technology, or to descriptions of external factors), and calculated, within each subgroup, the mean proportion of first order codes contained in the five overarching categories. This enabled descriptive between-groups comparison.

**Broad scoping literature review.** Near the end of the interview analysis (summer-autumn 2023), we reviewed conceptual and empirical academic papers discussing ethical issues related to the use of IoTs in OSH contexts. Following the PRISMA-ScR checklist [27], our goal was to list and describe the main opportunities and issues as discussed in the academic domain (detailed method and results are submitted in a separate paper). We chose this timing and procedure to minimize the theoretical influence of academic discussions on our empirical analysis. We then compared the two sources of information (interviewed stakeholders and academic literature), and searched for similarities, differences and blind spots. The main findings are commented in the discussion section.

## Results

The focus group included 7 participants. Due to a snowstorm, 3 participants joined late and missed the discussion of the connected chairs scenario (Posture-tracker Scen). We conducted 17 further individual online interviews. We reached data saturation before fulfilling our aimed variation sampling (total n = 24): coding of the last 7 interviews generated few new codes (88 out of 2202 in total) and none of these codes provided substantive added content.

Our sample, described in Table 1, includes 5 women and 19 men, which is reflective of the representation of women in the field of the Internet of Things technology [28, 29] with an average age range of 40–50. The focus group discussion lasted 3h30, and the individual interviews lasted between 45 and 80 min (60 min on average).

### Answers to closed questions

As described in the last column of Table 2, most participants (78%) found our scenarios realistic, and in 42% of the cases, they reported having seen or heard of similar technologies being used in Switzerland (mostly step-tracker watches). Interestingly, we observed some hesitation

of participants who could not firmly answer whether the cases they knew were comparable to our fictional scenarios, either because the device was the same but deployed with different goals (e.g. explicit tracing), or because the health purpose was the same, but the technology deployed was somewhat different.

In 19% of the cases, participants acknowledged not knowing enough to assess the legality of the scenarios. Except for the sound-tracker that was generally evaluated as illegal, a clear majority of participants evaluated that a deployment of the devices described in our scenarios is legal in Switzerland. Interestingly, participants tended to find all three scenarios more realistic than legally acceptable, meaning that they considered illegal use a true possibility.

A vast majority of participants reported that if they were the head of the company or an employee, they would reject the implementation of the sound-tracker. However, they reported more acceptance of the posture-tracker chairs or the step-tracker watches, especially if placed in the role of the head of the company.

Finally, a few participants explicitly changed their views during the interview, but no clear pattern can be observed.

### Answers to main open questions

Overall, while evaluating the advantages and ethical issues related to the three scenarios, participants expressed numerous ideas, generating 2202 codes that we grouped into five overarching categories: goal relevance, adverse side effects, role of employees, data process, and vagueness. The full book of codes is available on the OSF platform of the project: osf.io/e97v2.

The main results are summarised in Tables 3–7. They include information about the number of participants that mentioned the codes (nb-participants), and about the overall number of occurrences of those codes (nb-occurrence). The tables also systematically differentiate between numbers of full results (*total including follow-up discussions*) and results that came up before we asked follow-up questions (total excluding follow-up discussions). More detailed descriptions of the results are provided in S2 Text.

### Sub-group analyses

The subgroup analysis revealed no major differences in participants' sensitivity to specific ethical issues when stratified by typology of participant. As illustrated in Fig 1, women, men, and participants with or without conflicts of interest gave similar weight to each of the five groups of ethical issues. From the figure, we can see that participants generated more input (number of codes) on topics related to the data process. The vagueness category is artificially small: numerous codes included in the four other categories could also have been classified here.

### Literature review

Detailed results are provided in a separate paper [21]. In brief, we analysed 61 articles that fulfilled the selection criteria. Next to opportunities, we extracted an unexpectedly furnished list of ethical issues grouped in five overarching categories: "surveillance and problematic data re-purposing", "difficulty to inform, consult, and obtain consent from employees", "suboptimal data management", and long lists of "unintended and unpredictable adverse effects", and "contextual factors that are conducive to ethical issues". The review also highlights a lack of global ethical awareness in the academic realm: most articles only discuss briefly a few issues and none of the more critical articles spot only subsections of the issues that we found.

**Table 3. Goal relevance.**

nb-participants = 24; nb-occurrences 618 (total excluding follow-up discussions)
*nb-participants = 24; nb-occurrences = 823 (total including follow-up discussions)*

| | |
|---|---|
| **Benefits of the device**<br>nb-part = 24; nb-occ = 286<br>*F-up: nb-part = 24; nb-occ = 353* | Primary and secondary benefits: |
| | > Positive effects at the company level: managers more aware of health issues, more efficient health and risk prevention, improved working conditions and equity among employees, better trust relationships between employees and employers |
| | > Benefits for employees: increased awareness and literacy among employees, better health and quality of life for employees, more motivation at work |
| | > Benefits for the company: improved team building, reduced absenteeism, improved work performance |
| | > General improvement of public health<br>--> *F-up: Interest of collected aggregated data for public health* |
| | Skepticism that the IoT can provide the expected health benefits: |
| | > IoT is an irrelevant means to achieve the OSH goal |
| | > Advantages are likely counterbalanced by problematic data sharing procedures that may not be accepted by workers |
| | > Difficulty of implementing safeguards without impeding the achievement of intended goals |
| **Adequacy of the technology**<br>nb-part = 23; nb-occ = 132<br>*F-up: nb-part = 24; nb-occ = 223* | Worries about adequacy: technology should respond to a clear occupational health need identified before its deployment and obtain a positive result in a risk-benefit analysis, BUT many participants doubt that this is easy to achieve (technology not effective, invasive, too costly, fast outdated) |
| | > *F-up: pervasive aspect of the technology; leads to the development of inequitable OSH programs (useful for some employees and the reverse for others); is inadequate because replaces trust relationships with machines, because it is illegal or leads to dual use or surveillance* |
| | > *F-up: for some participants, the technology may be adequate if it responds to strong managerial needs* |
| | Once deployed, efficacy and accuracy of devices should be controlled (with validation tests, collection of evidence), BUT it is difficult to obtain quality tests based on precise data, and to interpret data correctly. Incentives for tests are low when the device is already commonly used |
| **Purposed dual use of the device** | Worries about dual use goals such as space management, the fight against absenteeism, productivity control that can lead to surveillance practices |
| nb-part = 23; nb-occ = 199<br>F-up: nb-part = 24; nb-occ = 245 | Some participants find that, in specific situations, this risk may be compensated by major security reasons, or common good purposes (e.g. organisational optimisation of working hours, improvement of algorithms) |
| **Limits for the acceptability of IoT for OSH goals** | Goals should be well explained to employees |
| *nb-part = 11; nb-occ = 25*<br>*F-up: nb-part = 15; nb-occ = 33* | Employees should have a right to oppose those goals |
| | Goals should not change |
| | Goals of data repurposing are not acceptable |
| | IoTs should not be deployed long-term |

## Discussion

Our study's findings are rich in content and shed light on how stakeholders react to an emerging realm of the digital workplace. In this section, we discuss our findings, highlighting the importance of one unexpected category (vagueness), link the empirical results of this research with our complementary literature review, and with similar empirical results.

First of all, it must be noted that participants were favourably disposed towards the use of some IoTs for OSH purposes (e.g. posture-tracker chair or step-tracker watch) while rejecting

**Table 4. Side effects.**

nb-participants = 24; nb-occurrences 251 (total excluding follow-up discussions)
*nb-participants = 24; nb-occurrences = 384 (total including follow-up discussions)*

| | |
|---|---|
| **Side effects for employees**<br>nb-part = 24; nb-occ = 208<br>*F-up: nb-part = 24; nb-occ = 312* | Risk of being the victim of errors (e.g. in case of low-quality data collection process, biased interpretation, algorithmic discrimination) |
| | Adverse effect in professional context: |
| | > Risk that employers re-purpose the data once at hand (even if not the primary intention) for problematic and discriminatory managerial decisions |
| | > Impaired work conditions: employees may be disturbed by the device while working; their workload or their work performance and may be unfavourably reevaluated; relationships between colleagues may by impaired by competitive comparisons |
| | > Gamification via IoTs may have the effect of under-evaluating the seriousness of OSH goals in users' mind, leading them to take more risks |
| | > *F-up: Invasive surveillance endangers employees' natural spontaneity or autonomous working habits (e.g. flexibility in working hours)* |
| | Adverse effects on the physical or psychological health of employees (e.g. due to stress of being monitored) |
| | Further unforeseen adverse effects: |
| | > Blurred limits between employees' professional and private life |
| | > Shift of health responsibilities from employers to employees because, once the device is deployed, employees seem to be in charge of following the health recommendations and maintaining good health |
| **Side effects for the companies and the society**<br><br>nb-part = 13; nb-occ = 47<br>*F-up: nb-part = 19; nb-occ = 84* | Surveillance practices involve the risk of damaging employer-employee trust relationships, and of impairing a company's public image: |
| | → *F-up: In reaction to surveillance, employees may develop new cheating habits that in turn may lead to increasingly invasive surveillance practices* |
| | → *F-up: increased use of IoTs make surveillance more common at the societal and work-levels and can lead to a dangerous pressure to conform and adopt 'healthy' behaviour* |
| | An increasing human-machine dependency could have adverse effects (e.g. lead to dehumanized workplace relationships; lead to less efficient OSH measures) |
| | IoT may be costly to maintain or deploy and necessitate additional organizational burdens |
| | *F-up: technology generates environmental issues* |

more privacy-invasive devices (e.g. sound-tracker on computer). This result is similar to previous results [22, 24].

Our findings show that participants pointed out an important number of ethical issues, which we grouped into five overarching categories. First, participants reflected on the *goal relevance* of the device. While acknowledging the most obvious OSH benefits provided by the technology, they expressed strong concerns about effectiveness and lack of clarity regarding employer goals, particularly problematic dual uses or later repurposing of data. Second, they were preoccupied with *adverse side effects*, for employees (e.g. psychological impact of monitoring, risk of algorithmic discrimination or distorted incentives and responsibilities), and for companies and society (e.g. risk of damaging trust in work relationships, of generating new cheating habits, of making surveillance more common). They emphasised the unpredictability of many adverse effects, making it difficult to set up efficient gatekeeping measures. Third, participants discussed the *role of employees* (the primary users of the devices), highlighting that their point of view in consultation processes and the quality of their consent were compromised by a number of factors: the complexity of the technology and data process, a lack of technological literacy and relationships of dependency. Participants emphasised that fair

**Table 5. Role of employees.**

nb-participants = 24; nb-occurrences 304 (total excluding follow-up discussions)
*nb-participants = 24; nb-occurrences = 403 (total including follow-up discussions)*

| | |
|---|---|
| **Freedom of choice** | Important to allow liberty and freedom of choice to employees |
| nb-part = 22; nb-occ = 80 *F-up: nb-part = 22; nb-occ = 113* | > But the fair possibility to opt out may not be given |
| | Risks associated with the right to choose: |
| | > If employees accept use of the device, discrimination and stigmatisation based on interpersonal comparisons may occur |
| | > Employees may be discriminated against if they refuse the device or do not master its use |
| **Information processing** nb-part = 13; nb-occ = 26 *F-up: nb-part = 14; nb-occ = 38* | Important to provide comprehensive information because it increases the ability to assess benefits, risks and ethical issues. But… |
| | > they need to receive clear and understandable information about the device or the data process |
| | > even if informed, they may not be supported to use the device in productive manner or to interpret the data in a useful way |
| | > *F-up: Risk that information leads employees to refuse IoTs for OSH purposes* |
| | Employees lack digital or technological literacy, making it difficult for them to use and assess them |
| **Employees' implication** nb-part = 23; nb-occ = 211 *F-up: nb-part = 23; nb-occ = 273* | Issues related to the involvement of employees in the decision process ahead of the deployment of the IoT |
| | > Consultation of employees is important because their feedback is helpful; it is a way to respect them and to guarantee their trust at work → *F-up: it favours employees' adherence to the device which is of practical use; transparency is an important value* |
| | > However, employers tend to avoid consultation because they fear employees' refusal |
| | > *F-up: Diverging views about whether all employees should be consulted in a direct democratic way or whether it is OK to discuss with knowledgeable representatives and set up safeguards* |
| | > *F-up: Limitations to consultations and information procedures: even if most employees are correctly informed and agree, there will always be some who disagree, or some residual mistrust* |
| | > *F-up: Some fatalistic considerations: employers will impose the devices on employees without consultation anyway* |
| | Problem of biases in the consultation process due to dependency relationships |
| | > Employees may not be free during the discussion process because of their subordinate status |
| | > Trust relationships depend on interpersonal factors (e.g. colleagues' attitude) |
| | > Engagement in the decision procedure may not be equally distributed among employees (e.g. consultation process may favour employees that are already healthy or take care of their health) |
| | Issues related to employees' consent |
| | > Important to seek consent especially when medical data is involved, but there are doubts about the value of the consent: |
| | > Employees lack critical vision because they are already habituated to the proliferation of technological and monitoring techniques |
| | > Employees' lack of freedom at the workplace decreases the value of their consent (due to direct pressure or unbalanced incentives) |
| | > Difficulty of obtaining transparent information from employers |
| | > Difficulty of gathering proper consent from all people while deploying the system (e.g. family members in a home office context) |
| | > Once they have consented, employees may face difficulties and challenges in exercising their rights to access or to control the processing of their data |
| | > Consent may be wrongly considered as the key solution to ethical issues, overshadowing other important difficulties listed above |

**Table 6. Data process.**

| | |
|---|---|
| nb-participants = 23; nb-occurrences = 274 (total excluding follow-up discussions) *nb-participants = 24; nb-occurrences = 784 (total including follow-up discussions)* | |
| **Specific issues related to various stages of data flow** nb-part = 22; nb-occ = 207 *F-up: nb-part = 24; nb-occ = 660* | Data collection |
| | > High volume of data collected involve higher risks of infringing the data minimization principle |
| | > Concern about the sensitivity of health and biometric data |
| | > Some participants consider data collection in the private sphere to be more acceptable than in a professional context (because in private contexts individuals are solely responsible for their choices) |
| | Data sharing |
| | > It is risky to share sensitive data with HR because HR may use the data for no obvious OSH purpose (conflict of interest towards employer/company); there is no obvious reason for HR to use the data for completing their tasks; F-up: HR do not have the knowledge and capacities to adequately interpret the data |
| | > It is risky to share sensitive data with Direct Managers (DM) → *F-up: Because DM lack the power to make decisions about organizational changes; DM are too close to employees who would not be free to refuse; it would create more control and intrusive knowledge about workers, which would stress employees; it would negatively affect working relationships between DM & employees and weaken the key role of DM as a link between employees and heads of company* |
| | > There are risks with sharing sensitive data with working colleagues |
| | > Concerns related to data sharing to third parties (e.g. external companies) → *F-up: because it is difficult to really anonymize the data* |
| | → *F-up: But some participants name exceptions: OK if data is anonymized and if the aim is to improve algorithms or the quality of the technology* |
| | > It is less problematic to share personal data with Occupational Physicians (OP). F-up: OP have more professional knowledge (they can detect poor-quality data and know how to interpret and use data in a meaningful way) and can develop global prevention measures, provide health recommendations and help optimize employees' personal health; OP can better help employees to interpret individual reports |
| | > It is less problematic when data is properly aggregated: according to some participants, in that case, even HR may access it |
| | Data storage |
| | > It is risky to store sensitive health data |
| | > Concern related to the location of the data storage: Switzerland is evaluated as a more secure place |
| | > *F-up: Some participants think that data should not be stored in the company: third-party actors (including GAFAM) can guarantee higher security standards against hackers* |
| | Data analysis |
| | > There are risks of dual use if the data analysis is done in-house (*F-up: related suspicion against HR); F-up: if it is done, it should be for health purposes only without any repurposing* |
| | > Externalized data analysis to a third-party company may be better; *F-up: a lack of internal competencies in the companies* |
| | > But worry about the risk of data reselling and breach of confidential information |
| | > *F-up: overall concern about data robustness (e.g. quality of source data) and quality of the analysis and interpretation (important contextual or individual characteristics may be missed)* |
| | > Concern about the transparency (e.g. how the analysis is done) |
| | > *F-up: It is better to analyze aggregated data* |

*(Continued)*

**Table 6.** (Continued)

| | |
|---|---|
| | Data governance |
| | > *F-up*: concern about lack of transparency (notably, unclear how the data are stored, managed, analyzed and transmitted in the general governance system of the data process) |
| | > *F-up*: too many intermediaries and actors are involved at different stages of the process; it is unclear what their role is |
| | > It is difficult to decide and clarify who is entitled and responsible for overall data management, and how this can be organized |
| **Data security**<br>nb-part = 16; nb-occ = 54<br>*F-up*: *nb-part = 19; nb-occ = 95* | Concern about the risk of data leaks, breaches and cyber-attacks |
| | More security is needed for medical data |
| | Risk that data are de-anonymized |
| | It is unclear under what process data are de facto accessible |
| **Lack of trustworthy actors**<br>nb-part = 8; nb-occ = 20<br>*F-up*: *nb-part = 17; nb-occ = 52* | There is a lack of trustworthy actors (e.g. competent authorities) who can regulate the data process |
| | > *F-up*: lack of nationally organized solutions, or local webs of trustworthy actors in the field of OSH |
| | > *F-up*: small companies have limited possibilities to offer high quality OSH services |
| | > *F-up*: not enough political will to enforce existing public institutions |
| | > *F-up*: as a result, data management and security is externalized to private actors or dependent on their competencies and willingness |
| | Occupational physicians (OPs) are trustworthy actors but not powerful enough: this is why they not consulted early enough in the decision process |
| | > *F-up*: OP are bound to medical secrecy thus more likely to use data in a responsible manner |
| | > *F-up*: However, there is the risk that OP are used by HR and end-up sharing sensitive data with their employer |

information, understanding of relevant issues and freedom of choice were difficult conditions to meet, undermining the relevance of consultation and consent procedures. Fourth, participants shared numerous concerns regarding the various stages of the *data process*, including data collection, transmission, storage, analysis, and overall governance. Most revolved around security and confidentiality issues; there was repetitively expressed distrust towards HR and frequent complaints about the lack of trustworthy actors, or existing trustworthy actors', such as occupational physicians, lack of power. They were insecure about their own capacity to identify the relevant facts (about the technology, the data flow, the status of the law, etc.) necessary for sound ethical evaluations.

Our last category, *Vagueness*, encompasses the aforementioned insecurity issue, as well as many further concerns. Participants expressed recurrent concerns about a lack of background knowledge and the intrinsic difficulty of obtaining relevant information for assessing where ethical problems lie. Vagueness issues occurred repetitively across all the other four topics. Participants expressed their lack of knowledge about multiple aspects that are important to understand in order to produce a sound ethical evaluation, such as how does the technology work; what is truly intended with its deployment; who gets access to the data; who knows (and controls) who gets access to the data; to what extent can we expect end-users to understand the relevant features of the technology and the consequences of its deployment; what are the long-term effects of the technology; what uses are legally acceptable; how can unavoidable conflict of interests be attenuated; what are employers' obligations; to what extent do IoTs fulfil true OSH goals, and what are the relevant methods and criteria for assessing such technology.

This vagueness issue is worrisome because withholding one's judgement is a wise attitude in the face of felt incompetence, but it can lead to a kind of ethical resignation, leaving

**Table 7. Vagueness.**

nb-participants = 24; nb-occurrences 169 total excluding follow-up discussions)
*nb-participants = 24; nb-occurrences = 210 (total including follow-up discussions)*

| | |
|---|---|
| **Unclear goals** | |
| nb-part = 20; nb-occ = 39 <br> *F-up: nb-part = 20; nb-occ = 40* | Employers may have unclear intentions (e.g. no clearly defined OSH goals) or base their decisions on unclear evidence of effectiveness |
| **Unclear employers' obligations** | |
| nb-part = 14; nb-occ = 37 <br> *F-up: nb-part = 16; nb-occ = 48* | Unclear what health promotion involves and what employers' obligations and responsibilities are |
| **Uncertainty around the evaluation of the technology** | Concerns about existing data and how (and whether it is possible) to evaluate the effectiveness and reliability of the technology |
| nb-part = 18; nb-occ = 57 <br> *F-up: nb-part = 20; nb-occ = 67* | Difficulty in evaluating if the devices fulfil OSH goals |
| | *> F-up: if an audit procedure takes place, it is unclear which criteria will be taken in consideration* |
| | Companies are overoptimistic leading to fast deployment without critical thinking, evaluation and safeguards |
| **Unclear legal issues** <br> nb-part = 11; nb-occ = 39 <br> *F-up: nb-part = 13; nb-occ = 58* | Law is unclear and interpretation is difficult |
| | *> International data transfer increases complexity of legal issues* |
| | *> F-up: difficult responsibility attribution of adverse effects* |
| | Unclear which device would fulfil OSH legal obligations |
| | Concern about the extent to which data protection law is or should be respected (e.g. how can employees exert ownership of their data? who and how should data be stored?) |
| | *> F-up: concerns about how to enforce compliance of devices with regulations, given the proliferation of new products?* |
| | *> F-up: could stronger regulation be a solution?* |

the floor open to all sorts of unregulated social experiments. Despite having been selected as the most knowledgeable people on IoTs, participants often felt unsure about what was is at stake in particular domains. For instance, in their answers to our closed questions, many of them confessed a lack of knowledge to assess the legality of our case scenarios, and those who did assess them expressed opposing opinions. Despite frequently expressing worries about lack of knowledge, the vast majority said that, in practice, they would endorse the deployment of an IoT such as the posture-tracker chair or the step-tracker watch, demonstrating either ethical naivety or resignation. In the domain of IoT for OSH goals, hardly anyone seems to feel fully competent, resulting in suboptimal ethical evaluation and a lack of precautionary or preventive measures. Such a result may indicate that stakeholders need to be supported by regulatory and political bodies that are responsible for supporting the ethical process.

Another interesting finding, linked to the vagueness issue, is the extent to which stakeholders miss relevant ethical thoughts if they are not actively informed of the details of the implementation procedure in real life. Participants in our study produced numerous novel insights once we showed them the data flow picture and asked follow-up questions. Although our follow-up inputs were purely descriptive (for methodological reasons, we did not want to introduce ethical themes during the interviews), they enabled participants to refine their analysis and further explore ethical aspects overlooked in the first place. We collected more thoughts on issues related to data quality, access, and governance, and on the rights, roles, and trustworthiness of stakeholders, including external third-party companies. The follow-up process also revealed divergences of opinion among stakeholders. They disagreed on whether consultation of employees should be done in direct and democratic ways or through

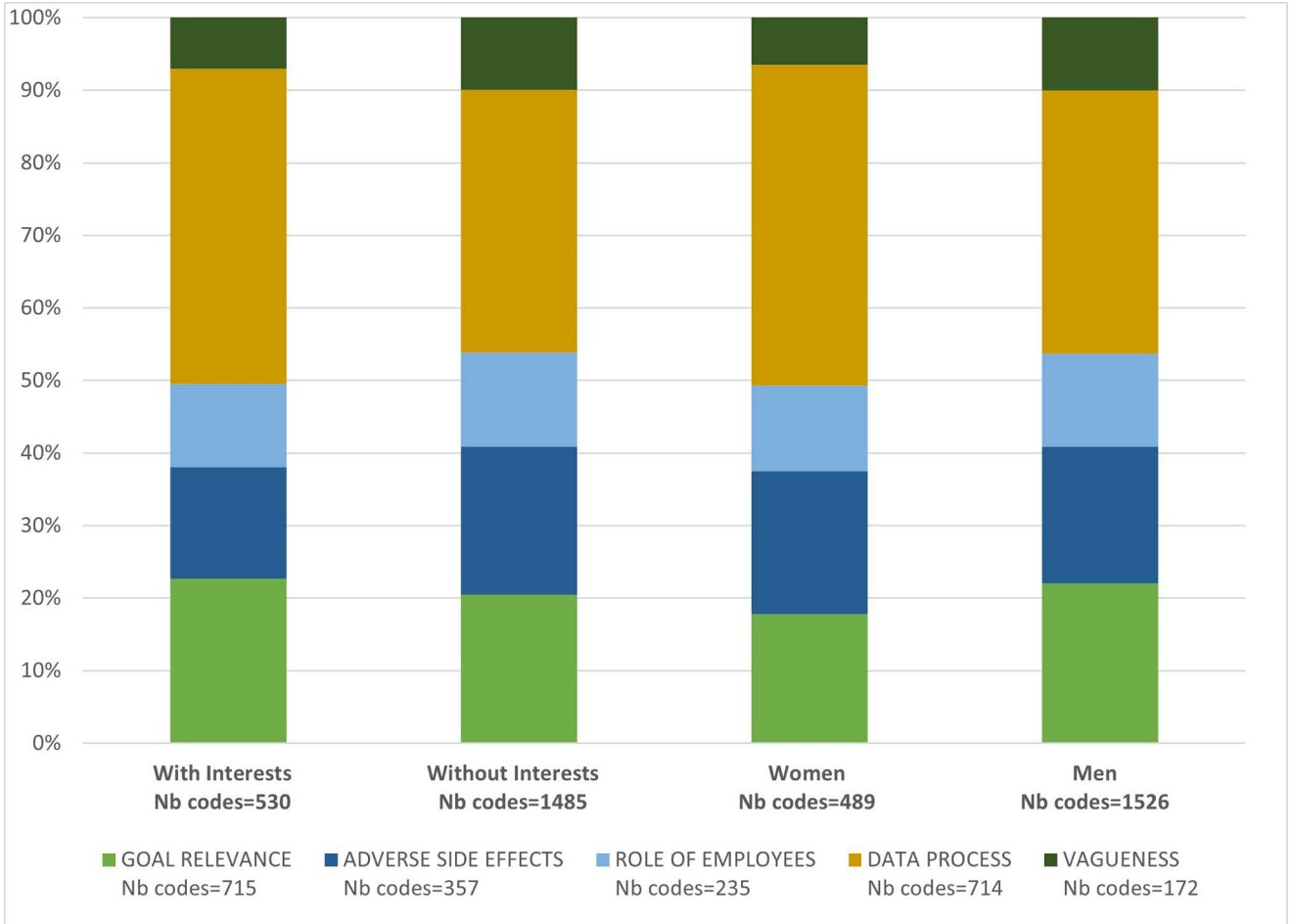

**Fig 1. Subgroup analysis - mean proportion of first order codes (related to ethical issues) contained in the five overarching categories (goal relevance; adverse side effects; role of employees; data process; vagueness) for each four sub-groups of participants: Women (n = 5), men (n = 19), participants with (n = 7) and without (n = 17) a conflict of interest.** Nb codes = total number of codes produced by all participants in the subgroup.

discussions with knowledgeable representatives. Some stakeholders objected to all forms of data repurposing, while others were ready to accept some repurposing upon sufficient justification.

Beyond this inquiry in two stages (questions and follow-up questions), participants' insights present interesting commonalities, differences, and blind spots with the topics discussed in the academic literature [21].

Overall, we found broad agreement among stakeholders, including those with or without conflicts of interest (as illustrated in Fig 1). The ethical issues they highlighted were fully in line with the worries expressed in the academic literature. Such a pattern is unfamiliar in practical ethical debates, often characterised by fierce fights between irreconcilable camps (think, for instance, of abortion, immigration, or environmental debates). This result engenders both hopes and worries. Hopes because effective solutions in communities can only be achieved through agreement on what matters and what should be improved. Worries because it is easy to agree on theoretical matters that have no practical effect whatsoever. An observed agreement may thus be a sign that we are far from any actionable

solution to address the ethical issues at stake (laws, regulations, guidelines, safeguard and mitigation measures).

Regarding differences with the academic literature, participants tended to express strong scepticism about the expected efficacy of IoT for OSH purposes and about the good intentions of employers who deploy them. They complained heavily about the lack of powerful trustworthy actors, making it difficult to enforce ethics-by-design procedures and mitigation measures. Participants expressed strong doubts about the applicability of fair consultation and consent procedures and, consequently, awarded them lesser value as an ethical means to protect employees. Such a resigned attitude was less noticeable in the academic literature [with exceptions e.g 16, 30].

Participants highlighted practical issues overlooked in the academic literature, such as the difficulty of establishing efficient safeguards without compromising the adequacy of the devices or the difficulty of defining employers' legal obligations. Further, they discussed at length the practical difficulties experienced by particular professions (HR, direct managers, OSH professionals, occupational physicians) when juggling a need to please employers while protecting employees, which are seldom thematised in the academic literature.

Reversely however, participants tended to overlook or to express only peripheral interest in numerous issues. The list of unintended and unpredictable adverse effects was strikingly shorter in stakeholders' reports. For instance, they hardly took seriously the risks of third-party exploitation of sensitive personal data that was a major concern in the literature [e.g 31–34], nor of workers' excessive trust in employers [e.g 35]. Overall, participants in our study did not take a global and holistic approach and rarely came up with societal and political level considerations. For instance, topics related to cyber criminality [e.g 36, 37], international surveillance [e.g 31–34], nor of workers' excessive trust in employers [e.g 35]. Overall, participants in our study did not take a global and holistic approach and rarely came up with societal and political level considerations. For instance, topics related to cyber criminality [e.g 36, 37], international surveillance [e.g 38–40], dystopian workplaces [e.g 32, 41], or social moves towards a banalisation of surveillance were missing [e.g 40, 42]. Similarly, participants hardly commented the role (and current limitations) of regulatory bodies and institutional safeguards which is heavily discussed in the literature [e.g 43–49].

Several factors may explain participants' stronger emphasis on subsets of ethical issues and their capacity to pinpoint specific issues that are overlooked in the academic literature. One is their first-hand experience that allows them to see vividly the problems that they have experienced, and, reversely, to overlook or express peripheral interest in issues that they have not experienced. We think that the risk that stakeholders underestimate some ethical issues because of a lack of personal experience is particularly high in the case of IoTs for the Good (i.e devices deployed for noble goals): the fact that those devices are deployed for OSH purposes tends to numb awareness of lurking ethical issues [e.g 16].

Another explanation of the observable differences between stakeholders' insights and the academic literature, is the cultural background in which the former are embedded. When discussing ethical issues, participants in our study considered work-context issues, taking place in the horizontal hierarchical relationships that characterise the classical internal organisation of Swiss companies. Except for general complaints about lacking laws or trustworthy actors, participants showed little interest in evaluating legal, administrative and institutional safeguards. The Swiss data protection authority (*préposé federal à la protection des données et à la transparence)*, which is supposed to play a crucial regulatory role, was not even mentioned. Such omission may reveal their low expectations of Swiss administrative and legal authorities. This is the mark of a strongly liberal culture within the working economy and public health.

Lastly, participants' selective ethical considerations may be partly due to methodological reasons: they were asked to discuss three specific case-scenarios that, for instance, did not include societal considerations.

Finally, our results complement existing research [22, 24, 35]. The method that we used (inclusion of a large variety of stakeholders; use of descriptive scenarios and open questions free from ethical concepts; explicit presentation of the data process in the scenario; open coding focusing on ethical *issues* and not on ethical *solutions*) allowed us to obtain a larger range of ethical insights without influencing content. We replicated most previous results: stakeholders are globally open-minded towards IoT for OSH purposes, but less open towards more intrusive devices. We observed a sensitivity to issues related to security, trust, consultation, side effects and intrusiveness. We obtained a richer set of concerns, notably related to the first-hand relevance of the technology, to various issues surrounding the data process, the lack of trustworthy actors and the overall lack of information enabling objective assessment of the technology. In contrast to Bovens, we did not find topics related to hierarchy and our stakeholders repetitively expressed a need for support to objectively evaluate the technology.

## Ethical highlights

This research provides important descriptive results that can inform ethical analysis. Here, we outline five ethical highlights that, in our view, deserve particular attention. Interestingly, they are all closely related to the vagueness issues revealed by our empirical analysis.

The discrepancy between the pace of technology and the pace of regulation increases the risk of exploiting vulnerable employees. As our results highlight, there is a lack of knowledgeable and trustworthy actors. Moreover, existing laws and guidelines for the safe deployment of IoTs are either lacking or unknown to relevant stakeholders (both in power and lower-hierarchy individuals). Meanwhile, technology and deployment options at the workplace and beyond are steadily increasing. Ways of exploiting subordinate employees are therefore extensive, varied, and difficult to trace. Examples include the harmful effects of peer-to-peer surveillance, the leakage of health data to third-party companies, or the use of employee health data for managerial purposes.

Inadequate ethical solutions are being used to protect employees. To address ethical issues raised by deploying IoT in the workplace, a commonly used solution is to organise consultation sessions and request consent from end-users. However, as is widely recognized in practical ethics, consent provided by vulnerable individuals can only validly justify the infringement of fundamental rights (such as bodily integrity or data privacy) when it is "free" and "informed" [50]. Our results show that these conditions are difficult to meet for IoT deployed for OSH purposes. The quality of end-users' consent may be impaired by multiple factors, including the complexity of the technology and data processes (partly entrusted to third parties), the technology's ability to be continually upgraded to include additional functionalities at various points, the overall lack of technological literacy among decision-makers and end-users, and dependency relationships in the workplace (e.g., employee subordination).

"Ethical overload" leads to "ethical resignation." Our data highlight two phenomena. First, most contributors (interviewed stakeholders, academic articles) seem only capable of considering a limited number of issues simultaneously when evaluating the acceptability of IoTs. Second, facing the complexity and vagueness issues, few feel fully competent to assess these devices, leading to reactions of helplessness and resignation. However, omission to help or to prevent adverse events is an inadequate or unacceptable ethical attitude.

There is a high risk of "ethical naivety" due to vagueness and complexity issues, coupled with the fact that IoTs for OSH purposes are viewed as IoTs for the Good. Stakeholders may

overlook or underestimate numerous ethical issues because they cannot clearly foresee or appreciate their significance (due to system complexity and a lack of personal experience with new devices), and because these IoTs are promoted with noble intentions (security or health benefits). Ethically naive stakeholders are vulnerable to manipulation through biased advertising campaigns created by powerful players who market these devices. The result is a lack of precautionary or preventive measures.

There is a risk of reinforcing a culture of surveillance. The deployment of IoTs for OSH purposes contributes to an overall trend of normalized surveillance in public and private spheres. In contemporary society, private and public tracing and monitoring devices (individual smartphones, smartwatches, cameras in workplaces and public spaces) pervade all areas of life. IoTs for OSH purposes are just another example of this trend. They therefore tend to escape ethical scrutiny and reinforce social acceptance of invasive surveillance practices.

## Regulatory recommendations

An in-depth legal and regulatory analysis of the legality of OSH tools in Switzerland is beyond the scope of this paper. However, our qualitative study has allowed us to highlight five specific ethical risks. Related to these, we make several recommendations for occupational health stakeholders.

To mitigate the risk of exploiting vulnerable employees, given the imbalance of power between employers and employees, authorities should increase the moral and legal responsibility of employers, rather than relying on consent solutions. Employers should be aware that they could be held civilly liable if an employee can prove a causal link between the use of IoTs and resulting harm. In this respect, the development and deployment of AI systems have altered the regulatory landscape. The European Union has decided to modify its liability regime, adopting a new EU directive on liability for Defective Products [51]. This new directive will be complemented by a second directive: the Artificial Intelligence Liability Directive [52]. Both legal texts aim to ensure that individuals are better protected in the digital age and to reverse the burden of proof. The first legislation targets manufacturers, while the second is still in the lawmaking process. However, they signal a shift in perspective that could influence Swiss law, especially if federal authorities choose to follow the European path in AI regulation. To increase employers' moral responsibility, OHS administrations and agencies should also inform employers and occupational health professionals about digital tools, their benefits and their risks. To this end, existing codes of ethics and professional training should be revised or developed.

To address the risks of ethical overload, ethical naivety, and reinforcing a surveillance culture, policymakers and legislators should explore a range of regulatory tools, combining impact assessments, regulatory sandboxes, transparency requirements, and training. Preliminary impact assessments should be conducted not only in relation to the protection of personal data but also in relation to other rights, such as the principle of non-discrimination and the right to privacy. Given the difficulty of anticipating the risks associated with rapidly evolving technology, an approach based on experimentation under the supervision of a regulatory authority would provide concrete, rather than hypothetical, knowledge of the effects. Preliminary impact analyses and experiments would provide more information about the effects of IoTs, helping to counter ethical naivety and ethical overload. Moreover, transparency measures would require documentation efforts during tool development and use. If negative impacts are identified in a context of use, documentation requirements will provide the information and data needed to understand and explain why these impacts occurred. In addition, a management-based approach should be deployed by OSH administration and

agencies to increase health professionals' awareness through training and information campaigns. Furthermore, all these measures and their outcomes should also be accessible to all, enabling workers advocacy organisations to act as watchdogs.

Nevertheless, all these measures will be in vain if they are not accompanied by obligations and control mechanisms. To this end, these measures should be enshrined in law, with enforcement monitored and penalties imposed for breaches. As has been observed in many areas (e.g., social media, sustainable development), a soft regulatory based on self-governance response is insufficient in cases of power imbalances and high economic stakes [53, 54]. The legislator must find the right combination to protect the weaker party without stifling technological solutions that could improve workplace health.

## Limitations

It is important when interpreting the results to remember, as with all qualitative research, that numbers reported in the paper signify salience rather than frequency and should not be generalized as quantitatively representative.

We recruited only within the Swiss working population. Participants living in different countries or with different cultural backgrounds may have shown different patterns of concerns. Awareness of ethical issues is tightly linked to external factors such as existing legal frameworks (which vary from one country to another) or past experiences of scandals covered by the local media.

Participants in this study had a high level of education, which may limit the generalization of our findings to people with lower levels of education. There were more male participants than female participants, which does not correspond to the gender spread of employees impacted by IoTs. Moreover, depending on personal features (e.g. shyness, capacity to elaborate on inputs in a discussion) we cannot exclude that some participants would have provided more insights if they had participated to an individual interview rather than the focus group, or reversely.

The three case-study scenarios used do not illustrate all features and consequences related to the use of IoTs for OSH purposes. They do not involve IoTs responding to short-term security risks, nor ill-intentioned actors or broad societal consequences. Such focus partly explains why participants failed to address all relevant ethical aspects or emphasised some aspects at the expense of other.

Since we aimed at an overall collection of ethical issues, while analysing the transcripts, we did not investigate whether and how participants change their ethical evaluation depending on specific features of the three scenarios. This limitation could be addressed with a complementary analysis on the transcripts.

By design, we asked our closed questions at the beginning of the interview, but did not explicitly check whether participants had changed their mind at the end of the interview. We only recorded participants' unprompted and explicitly expressed changes of mind at the coding stage. In afterthought, it would have been interesting to ask the same closed questions at the end of the interview to systematically record participants' changes of mind.

## Conclusions

Our study provides novel insights into the perspectives of relevant stakeholders regarding the digitalization of the workplace and the implementation of IoTs for OSH purposes. The numerous ethical issues highlighted here and in our connected literature review [21] indicate that even if deployed for OSH purposes, IoTs can easily lead to irresponsible use. Stakeholders'

inputs were partly disenchanted and shed a crude light on the lack of information at their disposal even while in the front line to assess such technology.

Increased knowledge about relevant ethical dimensions is a necessary first step to foster responsible use of this technology in the workplace. Future research should aim to address some of our limitations by including a more culturally diverse sample of participants and varying the case scenarios. Nevertheless, our results provide important ground material, crossing ethical fields (business, health and digital ethics), on which to base necessary and still awaited ethical frameworks for assessing such technology, guidelines and regulation instruments.

## Supporting information

**S1 Text. Interview grids.**
(DOCX)

**S2 Text. Full description of qualitative results.**
(DOCX)

## Acknowledgments

We thank Marie-Gabrielle Mansour for her contribution to the design and focus group, Dana Naous for her contribution to the design and double coding.

## Author contributions

**Conceptualization:** M. El Bouchikhi, Sophie Weerts, Christine Clavien.

**Data curation:** M. El Bouchikhi, Christine Clavien.

**Formal analysis:** M. El Bouchikhi, Sophie Weerts, Christine Clavien.

**Funding acquisition:** Sophie Weerts, Christine Clavien.

**Investigation:** M. El Bouchikhi, Sophie Weerts, Christine Clavien.

**Methodology:** M. El Bouchikhi, Sophie Weerts, Christine Clavien.

**Project administration:** Sophie Weerts, Christine Clavien.

**Supervision:** Sophie Weerts, Christine Clavien.

**Validation:** Sophie Weerts, Christine Clavien.

**Visualization:** Sophie Weerts, Christine Clavien.

**Writing – original draft:** M. El Bouchikhi, Sophie Weerts, Christine Clavien.

**Writing – review & editing:** M. El Bouchikhi, Sophie Weerts, Christine Clavien.

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
