## [Decision Letter · Decision Letter 0]

1 Oct 2024

PONE-D-24-31290The Internet of Things Deployed for Occupational Health and Safety Purposes: A Qualitative Study of Opportunities and Ethical IssuesPLOS ONE

Dear Dr. Clavien,

Thank you for submitting your manuscript to PLOS ONE. After careful consideration, we feel that it has merit but does not fully meet PLOS ONE’s publication criteria as it currently stands. Therefore, we invite you to submit a revised version of the manuscript that addresses the points raised during the review process.

The two reviewers both commend the paper for its timely focus on the ethical concerns of IoT in occupational safety and health, but also identify areas where the paper could be strengthened. One reviewer suggests that the paper could provide more detail about the literature review and differentiate participant responses based on the scenarios. The other reviewer emphasizes the need for more information on stakeholder selection, deeper ethical analysis, and concrete policy recommendations. The paper is to be improved by addressing these specific areas.

We look forward to receiving your revised manuscript.

Kind regards,

Denis Alves Coelho, PhD

Academic Editor

PLOS ONE

Journal Requirements:

 SW and CC obtrainded funding for this research from the Swiss National Science Foundation (grant no. 187429) within the Swiss National Research Programme (NRP77) on “Digital Transformation” (https://www.snf.ch/en/hRMuYd5Qqjpl1goQ/page/researchinFocus/nrp/nrp77). The SNSF is a nonprofit national funding agency that was not involved in any step of this study. The authors have no financial or competitive interests to declare.  

5. Please include captions for your Supporting Information files at the end of your manuscript, and update any in-text citations to match accordingly. Please see our Supporting Information guidelines for more information: http://journals.plos.org/plosone/s/supporting-information .

Additional Editor Comments :

Reviewer 1 finds the paper to be a valuable contribution but suggests improvements in stakeholder selection, ethical analysis, and policy recommendations. Reviewer 2 recommends strengthening the paper by providing more context, addressing methodological questions, and clarifying specific points.

Reviewers' comments:

Reviewer's Responses to Questions

**Comments to the Author**

1. Is the manuscript technically sound, and do the data support the conclusions?

Reviewer #1: Yes

Reviewer #2: Yes

2. Has the statistical analysis been performed appropriately and rigorously?

Reviewer #1: Yes

Reviewer #2: Yes

3. Have the authors made all data underlying the findings in their manuscript fully available?

Reviewer #1: Yes

Reviewer #2: No

4. Is the manuscript presented in an intelligible fashion and written in standard English?

Reviewer #1: Yes

Reviewer #2: Yes

5. Review Comments to the Author

Reviewer #1: This paper examines the ethical concerns related to the deployment of Internet of Things (IoT) technologies in occupational safety and health (OSH) settings. The authors rightly point out the increasing use of IoT in various sectors, including workplaces, but identify a significant gap in the literature regarding the ethical implications. Through a qualitative approach involving focus groups and semi-structured interviews with 24 stakeholders, the study explores their attitudes toward different IoT devices and the ethical challenges they raise.

The methodology employed, including the use of open coding for transcript analysis, is appropriate for this type of exploratory research. The stakeholders’ perspectives revealed nuanced views on different types of IoT devices. They supported some, like posture-tracker chairs and step-tracker watches, but expressed concern over others, such as sound-tracking devices. The ethical concerns were grouped into five categories: goal relevance, adverse side effects, employee roles, data processes, and vagueness, which aligns with concerns raised in academic discussions on similar technologies.

However, a few areas of the paper could be strengthened:

Clarity on Stakeholder Selection: The paper would benefit from more details on how stakeholders were selected for the study. Were they from diverse industries, or were they primarily from sectors heavily reliant on IoT technologies? A more thorough explanation of their background could provide better context for the results.

Depth of Ethical Analysis: While the categorization of ethical concerns is helpful, the analysis could delve deeper into each category. For instance, the "role of employees" is briefly mentioned, but further discussion on how IoT might impact worker autonomy, consent, and surveillance could enhance the ethical argument.

Regulatory Recommendations: The conclusion mentions the need for guidelines and regulatory instruments, but the paper stops short of offering concrete recommendations. Including more specific policy suggestions or frameworks for future regulation would significantly improve the paper's practical relevance.

Overall, the paper is an important contribution to the growing discourse on the ethical challenges of IoT in workplace settings. It offers valuable stakeholder insights but could improve by providing deeper ethical analysis and clearer policy recommendations.

Reviewer #2: Overall, I find the paper to be well structured and written, clearly motivated and reporting relevant empirical findings. The obtained data support the drawn conclusions, and the reported analyses are appropriate for the type of data collected.

In terms of data availability, the authors have done a good job of providing an easily accessible summary of the qualitative data in the supporting materials. However, the authors have not provided the (anonymized) transcripts of the focus group and interviews. The authors should indicate why this is not possible or add the transcripts to the data repository.

For suggestions for further improvement, see my comments below.

L90 My main concern with the manuscript is the ability of the reader to follow parts of the presented argument without having more insight into the separate literature review. I understand that the combination of the two studies may be beyond the scope of a single paper. However, I would suggest adding more detail about the literature review to make this paper (especially the discussion section) read better on its own. This should include information on the number of reviewed articles as well as the types of articles - empirical vs. conceptual.

L137 You list a number of aspects that differed between the three examined scenarios. Given this description, I would have expected an analysis that differentiated participants' responses based on these factors. I understand that the aim of this article was the overall collection of ethical issues rather than differentiating between the scenarios. I would suggest mentioning this in the limitations section.

L167 Was there a systematic process for identifying the different stakeholder groups?

L173 What was the rationale for conducting a focus group rather than additional individual interviews? You should touch on the methodological differences between the two approaches and the potential for different results (or lack thereof) in the discussion or limitation section.

L208 Typo in “spitted”

L243 The statement on the representation of women in the field of the Internet of Things technology requires a reference

L241 Why 18 or 19? How was this judged?

L245 It is a bit confusing why the duration of the focus group discussion is specified per scenario but the duration of the interviews (seemingly) for all scenarios combined.

L332-L342 How did the interviewer respond to the participants' questions? The level of detail in which the scenarios were prepared should be discussed in terms of its relevance to the obtained responses.

L382 “They overlooked the risk of deskilling workers when their tasks are progressively replaced by the technology” – None of the three examined scenarios include a technology that would replace relevant human work tasks. This highlights that there are likely to be differences in the types of technologies considered in this study and included in the literature review. This aspect should be addressed by providing relevant information on the literature base of the review and/or by appropriate discussion.

6. PLOS authors have the option to publish the peer review history of their article (what does this mean?). If published, this will include your full peer review and any attached files.

**Do you want your identity to be public for this peer review?** For information about this choice, including consent withdrawal, please see our Privacy Policy .

Reviewer #1: No

Reviewer #2: No

---

## [Author Response · Author response to Decision Letter 0]

15 Nov 2024

Thanks for allowing us to resubmit a revised version of our article. The comments we received from the reviewers were very helpful. We have addressed them all in this version and hope that it now meets your publication criteria.

Comments & Responses to academic editor

Comment 1: Please ensure that your manuscript meets PLOS ONE's style requirements, including those for file naming.

Response: Done

Comment 2: Please provide additional details regarding participant consent. In the ethics statement in the Methods and online submission information, please ensure that you have specified what type you obtained (for instance, written or verbal, and if verbal, how it was documented and witnessed). If your study included minors, state whether you obtained consent from parents or guardians. If the need for consent was waived by the ethics committee, please include this information.

Response: We have made it more explicit that we obtained written consent from participants.

Comment 3: Thank you for stating the following financial disclosure: SW and CC obtainded funding for this research from the Swiss National Science Foundation (grant no. 187429) within the Swiss National Research Programme (NRP77) on “Digital Transformation” (https://www.snf.ch/en/hRMuYd5Qqjpl1goQ/page/researchinFocus/nrp/nrp77). The SNSF is a nonprofit national funding agency that was not involved in any step of this study. The authors have no financial or competitive interests to declare. Please state what role the funders took in the study. If the funders had no role, please state: ""The funders had no role in study design, data collection and analysis, decision to publish, or preparation of the manuscript."" If this statement is not correct you must amend it as needed. Please include this amended Role of Funder statement in your cover letter; we will change the online submission form on your behalf.

Response: Here is an updated wording of the §, including the amended role of funder statement: SW and CC obtained funding for this research from the Swiss National Science Foundation (grant no. 187429) within the Swiss National Research Programme (NRP77) on “Digital Transformation” (https://www.snf.ch/en/hRMuYd5Qqjpl1goQ/page/researchinFocus/nrp/nrp77). The funders had no role in study design, data collection and analysis, decision to publish, or preparation of the manuscript. The authors have no financial or competitive interests to declare.

Comment 4: Please include your full ethics statement in the ‘Methods’ section of your manuscript file. In your statement, please include the full name of the IRB or ethics committee who approved or waived your study, as well as whether or not you obtained informed written or verbal consent. If consent was waived for your study, please include this information in your statement as well.

Response: Done

Comment 5: Please include captions for your Supporting Information files at the end of your manuscript, and update any in-text citations to match accordingly.

Response: Done

Comment 6: Please review your reference list to ensure that it is complete and correct. If you have cited papers that have been retracted, please include the rationale for doing so in the manuscript text, or remove these references and replace them with relevant current references. Any changes to the reference list should be mentioned in the rebuttal letter that accompanies your revised manuscript. If you need to cite a retracted article, indicate the article’s retracted status in the References list and also include a citation and full reference for the retraction notice.

Response: We checked. None of the articles cited in our reference list was retracted. We have also added several articles to the reference list:

- Our literature review (tightly linked to this paper) has been accepted for publication this week and can now be cited.

- We added one methodological reference (on the PRISMA checklist) because reviewer 2 (comment 2) asked to provide more details about our connected literature review.

- We added one reference to support our statement on the representation of women in the field of the Internet of Things technology (asked by reviewer 2 in comment 7)

- Several new references have been added in the new sub-sections “ethical highlights” and “regulatory recommendations” (asked by reviewer 1, comments 2&3)

Comment 7: Reviewer 1 finds the paper to be a valuable contribution but suggests improvements in stakeholder selection, ethical analysis, and policy recommendations. Reviewer 2 recommends strengthening the paper by providing more context, addressing methodological questions, and clarifying specific points.

Response: We have made all the suggested modifications and additions. See detailed responses to reviewers below.

Comment 8: Have the authors made all data underlying the findings in their manuscript fully available?

Reviewer #1: Yes

Reviewer #2: No

Response: Reviewer 2 asks whether it is possible to upload the transcripts on the repository. Unfortunately, we cannot do so. We explain this in a new § in the “data availability” section.

“Some of our participants are public figures and some inputs during the discussions provide sensitive information about current practices in recognizable local companies. Due to the risk of de-anonymisation, we committed not to make transcripts publicly available in the data repository.”

Comments & Responses to reviewer 1

Comment 1: Clarity on Stakeholder Selection: The paper would benefit from more details on how stakeholders were selected for the study. Were they from diverse industries, or were they primarily from sectors heavily reliant on IoT technologies? A more thorough explanation of their background could provide better context for the results.

Response: Thanks for the comment. In fact, we aimed for and achieved a large diversity of stakeholders and should have explained it better in the paper. We now have modified the main § in the “participant recruitment” section as follow: “Our goal was to include a broad range of actors in different domains of activity related to IoTs: we aimed at covering knowledge and competencies related to 1) the development and production of IoTs (e.g. IT developer, manager in a start-up producing IoTs), 2) the deployment and use of IoTs in private companies (e.g. private company manager, Human Resources manager, OSH professional, occupational physician, end-user employees, advocate of employees, health insurance representative), and 3) the assessment, control and regulation of such use (e.g. lawyer, public administrator, cantonal physician, tech journalist, critical analyst working in a university or think tank). Overall, we also aimed at a diversity in gender, hierarchical level on the workplace, socio-cultural regions in Switzerland, and a variety of public and private working sectors (e.g. university, high tech start-up, federal administration, think-tank, media sector, goods-producing company, selling company, politics, health insurance).”

Comment 2: Depth of Ethical Analysis: While the categorization of ethical concerns is helpful, the analysis could delve deeper into each category. For instance, the "role of employees" is briefly mentioned, but further discussion on how IoT might impact worker autonomy, consent, and surveillance could enhance the ethical argument.

Response: Following this comment, we have added an “ethical highlights” section at the end of the discussion.

Comment 3: Regulatory Recommendations: The conclusion mentions the need for guidelines and regulatory instruments, but the paper stops short of offering concrete recommendations. Including more specific policy suggestions or frameworks for future regulation would significantly improve the paper's practical relevance.

Response: In the same line, we have added a “regulatory recommendations” section at the end of the discussion.

Comment 4: Overall, the paper is an important contribution to the growing discourse on the ethical challenges of IoT in workplace settings. It offers valuable stakeholder insights but could improve by providing deeper ethical analysis and clearer policy recommendations.

Response: Thanks for your comments and for inviting us to dig more into the ethical and recommendation aspects. We took that opportunity to expand the discussion by adding two new sections at the end of the article.

Comments & Responses to reviewer 2

Comment 1: the authors have not provided the (anonymized) transcripts of the focus group and interviews. The authors should indicate why this is not possible or add the transcripts to the data repository.

Response: Thanks for the comments. We have added the following § to explain why the transcripts are not available on the data repository: “Some of our participants are public figures and some inputs during the discussions provide sensitive information about current practices in recognizable local companies. Due to the risk of de-anonymisation, we committed not to make transcripts publicly available in the data repository.”

Comment 2: L90. My main concern with the manuscript is the ability of the reader to follow parts of the presented argument without having more insight into the separate literature review. I understand that the combination of the two studies may be beyond the scope of a single paper. However, I would suggest adding more detail about the literature review to make this paper (especially the discussion section) read better on its own. This should include information on the number of reviewed articles as well as the types of articles - empirical vs. conceptual.

Response: Following your suggestion, we have expanded the description of our literature review (which has meanwhile been accepted for publication) in providing more details in the method section and in the result section. Following the PRISMA-ScR checklist, our goal was to list and describe the main opportunities and issues as discussed in the academic domain. We analysed 61 articles that fulfilled the selection criteria. Next to opportunities, we extracted an unexpectedly furnished list of ethical issues grouped in five overarching categories: “surveillance and problematic data re-purposing”, “difficulty to inform, consult, and obtain consent from employees”, “suboptimal data management”, and long lists of “unintended and unpredictable adverse effects”, and “contextual factors that are conducive to ethical issues”. The review also highlights a lack of global ethical awareness in the academic realm: most articles only discuss briefly a few issues and none of the more critical articles spot only subsections of the issues that we found.

Comment 3: L137 You list a number of aspects that differed between the three examined scenarios. Given this description, I would have expected an analysis that differentiated participants' responses based on these factors. I understand that the aim of this article was the overall collection of ethical issues rather than differentiating between the scenarios. I would suggest mentioning this in the limitations section.

Response: Done. We have added the following sentence in the limitation section: “Since we aimed at an overall collection of ethical issues, while analysing the transcripts, we did not investigate whether and how participants change their ethical evaluation depending on specific features of the three scenarios. This limitation could be addressed with a complementary analysis on the transcripts.

Comment 4: L167 Was there a systematic process for identifying the different stakeholder groups?

Response: Reviewer 1 made a similar comment. We had diversity criteria in mind and should have provided more details in the article. We have now amended the main § in the “participant recruitment” section.

Comment 5: L173 What was the rationale for conducting a focus group rather than additional individual interviews? You should touch on the methodological differences between the two approaches and the potential for different results (or lack thereof) in the discussion or limitation section.

Response: One rationale for the focus group/interview was explained in the “data collection” subsection, but we agree that it may not be the best place to locate that information. We have now moved and expanded this paragraph in the first § of the Method section: “We chose this double method because focus groups usually enable debate, discussion, sharing of ideas, and generate more refined lines of thought, while individual interviews provide more space for shyer participants to speak, especially for participants from lower down in the hierarchy. Interviews also allow to adapt the language (French, English) to participants’ ease.”.

In the limitation section, we have now added the following sentence: “Depending on personal features (e.g. shyness, capacity to elaborate on inputs during a discussion) we cannot exclude that some participants would have provided more insights if they had participated to an individual interview rather than the focus group, or reversely.”

Comment 6: L208 Typo in “spitted”

Response: Thanks for noticing it! It is corrected.

Comment 7: L243 The statement on the representation of women in the field of the Internet of Things technology requires a reference

Response: As supporting reference, we have added an UNESCO report that provides international data on the lack of women in STEM disciplines.

Comment 8: L241 Why 18 or 19? How was this judged?

Response: In fact, this was a general impression we got while coding, which we did not feel necessary to substantiate since we had to add 5 or 6 more interviews until we reached the aimed sampling variation. We have now conducted a post-hoc analysis by sequentially subtracting codes of the last interviews and provide the results of this analysis in the first § of the Results section: “coding of the last 7 interviews generated few new codes (88 out of 2202 in total) and none of these codes provided substantive added content”.

Comment 9: L245 It is a bit confusing why the duration of the focus group discussion is specified per scenario but the duration of the interviews (seemingly) for all scenarios combined.

Response: Indeed: we simplified the sentence and now provide the total time for the focus group and the interviews: “The focus group discussion lasted 3h30, and the individual interviews lasted between 45 and 80 min (60 min on average)”.

Comment 10: L332-L342 How did the interviewer respond to the participants' questions? The level of detail in which the scenarios were prepared should be discussed in terms of its relevance to the obtained responses.

Response: For methodological reasons, the interviewers did not respond to participants’ questions, except for understanding questions. But mostly these questions were not explicitly asked to interviewers. They were rhetorical or came up within deliberative reasoning while participants discussed with each other (focus group) or expressed their thoughts. This said, your comment highlights that our sentence is formulated in a misleading way. We have now reformulated the § by removing the question marks style and explaining that “participants expressed their lack of knowledge about multiple aspects that are important to understand in order to produce a sound ethical evaluation”.

Comment 11: L382 “They overlooked the risk of deskilling workers when their tasks are progressively replaced by the technology” – None of the three examined scenarios include a technology that would replace relevant human work tasks. This highlights that there are likely to be differences in the types of technologies considered in this study and included in the literature review. This aspect should be addressed by providing relevant information on the literature base of the review and/or by appropriate discussion.

Response: Indeed and thanks for the comment. We have removed this sentence and acknowledge in the limitation section that our scenarios “do not involve IoTs responding to short-term security risks”.

---

## [Decision Letter · Decision Letter 1]

29 Nov 2024

The internet of things deployed for occupational health and safety purposes: A qualitative study of opportunities and ethical issues

PONE-D-24-31290R1

Dear Dr. Clavien,

We’re pleased to inform you that your manuscript has been judged scientifically suitable for publication and will be formally accepted for publication once it meets all outstanding technical requirements.

An invoice will be generated when your article is formally accepted. Please note, if your institution has a publishing partnership with PLOS and your article meets the relevant criteria, all or part of your publication costs will be covered. Please make sure your user information is up-to-date by logging into Editorial Manager at Editorial Manager®  and clicking the ‘Update My Information' link at the top of the page. If you have any questions relating to publication charges, please contact our Author Billing department directly at authorbilling@plos.org .

If your institution or institutions have a press office, please notify them about your upcoming paper to help maximize its impact. If they’ll be preparing press materials, please inform our press team as soon as possible -- no later than 48 hours after receiving the formal acceptance. Your manuscript will remain under strict press embargo until 2 pm Eastern Time on the date of publication. For more information, please contact onepress@plos.org .

Kind regards,

Denis Alves Coelho, PhD

Academic Editor

PLOS ONE

Additional Editor Comments (optional):

Reviewers' comments:

Reviewer's Responses to Questions

**Comments to the Author**

1. If the authors have adequately addressed your comments raised in a previous round of review and you feel that this manuscript is now acceptable for publication, you may indicate that here to bypass the “Comments to the Author” section, enter your conflict of interest statement in the “Confidential to Editor” section, and submit your "Accept" recommendation.

Reviewer #1: All comments have been addressed

Reviewer #2: All comments have been addressed

2. Is the manuscript technically sound, and do the data support the conclusions?

Reviewer #1: Yes

Reviewer #2: Yes

3. Has the statistical analysis been performed appropriately and rigorously?

Reviewer #1: Yes

Reviewer #2: Yes

4. Have the authors made all data underlying the findings in their manuscript fully available?

Reviewer #1: Yes

Reviewer #2: Yes

5. Is the manuscript presented in an intelligible fashion and written in standard English?

Reviewer #1: Yes

Reviewer #2: Yes

6. Review Comments to the Author

Reviewer #1: 

What is the scientific significance of this work?

Reviewer #2: I congratulate the authors on their good work and well-crafted manuscript. I have no further comments.

7. PLOS authors have the option to publish the peer review history of their article (what does this mean?). If published, this will include your full peer review and any attached files.

**Do you want your identity to be public for this peer review?** For information about this choice, including consent withdrawal, please see our Privacy Policy .

Reviewer #1: No

Reviewer #2: No

---

## [Editor Report · Acceptance letter]

5 Dec 2024

PONE-D-24-31290R1

PLOS ONE

Dear Dr. Clavien,

I'm pleased to inform you that your manuscript has been deemed suitable for publication in PLOS ONE. Congratulations! Your manuscript is now being handed over to our production team.

Lastly, if your institution or institutions have a press office, please let them know about your upcoming paper now to help maximize its impact. If they'll be preparing press materials, please inform our press team within the next 48 hours. Your manuscript will remain under strict press embargo until 2 pm Eastern Time on the date of publication. For more information, please contact onepress@plos.org .

If we can help with anything else, please email us at customercare@plos.org .

Kind regards,

on behalf of

Dr. Denis Alves Coelho

Academic Editor

PLOS ONE